

# Inpainting Radar Missing Data Regions with Deep Learning

Andrew Geiss and Joseph C. Hardin

Pacific Northwest National Laboratory, Richland, WA, USA

**Correspondence:** Andrew Geiss (andrew.geiss@pnnl.gov)

**Abstract.** Missing and low-quality data regions are a frequent problem for weather radars. They can stem from a variety of sources: beam blockage, instrument failure, near-ground blind zones, and many others. Filling-in missing data regions is often useful for estimating local atmospheric properties and application of high-level data processing schemes without the need for preprocessing and error-handling steps; feature detection and tracking for instance. Interpolation schemes are typically used for this task, though they tend to produce unrealistically spatially-smoothed results that are not representative of the atmospheric turbulence and variability that are usually resolved by weather radars. Recently, Generative Adversarial Networks (GANs) have achieved impressive results in the area of photo inpainting. Here, they are demonstrated as a tool for infilling radar missing data regions. These neural networks are capable of extending large-scale cloud and precipitation features that border missing data regions into the regions while hallucinating plausible small scale variability. In other words, they can inpaint missing data with accurate large-scale features and plausible local small-scale features. This method is demonstrated on a scanning C-band and vertically pointing Ka-band radar that were deployed as part of the Cloud Aerosol and Complex Terrain Interactions (CACTI) field campaign. Three missing data scenarios are explored: infilling low-level blind zones and short outage periods for the Ka-band radar, and infilling beam blockage areas for the C-band radar. Two deep learning based approaches are tested, a Convolutional Neural Network (CNN) and a GAN that optimize pixel level error or combined pixel level error and adversarial loss respectively. Both deep learning approaches significantly outperform traditional inpainting schemes under several pixel-level and perceptual quality metrics.

## 1 Introduction

Missing data regions are a common problem for weather radars and can arise for many reasons. One of the most common for scanning radars is beam blockage. This occurs when terrain or nearby objects like buildings and trees obstruct the radar beam, resulting in a wedge-shaped blind zone behind the object. This is a particularly large problem in regions with substantial terrain like the western potion of the United States for instance (Westrick et al., 1999; Young et al., 1999). Scanning radars can suffer from many other data quality issues where contiguous regions of missing or low-quality data may need to be inpainted. Some examples are: interference from solar radiation at dawn and dusk (Liu et al., 2016), ground clutter (Hubbert et al., 2009, 01



Jul. 2009) and super-refraction (Moszkowicz et al., 1994), and echos off of wind farms (Isom et al., 2009). Complete beam
extinction due to attenuation by large storms can similarly cause large missing data regions for high frequency radars. Several
computational approaches exist to infill partial beam blockage cases (Lang et al., 2009; Zhang et al., 2013). Another option is
to use a radar network where multiple radars are installed on opposite sides of terrain (Young et al., 1999). More recently, deep-
learning-based data fusion techniques (Veillette et al., 2018) have been developed to enhance the coverage of radar networks
by emulating radar observations based on data from satellite imagers and other instruments; these techniques have promise
for combating beam blockage and large missing data regions for scanning radars. Finally, in the absence of additional data
(from unblocked sweeps at higher elevation angles or other instruments), beam blockages can be filled in through traditional
interpolation.

In addition to beam blockage for scanning radars, we also examine simulated missing data scenarios for vertically pointing
radars. Specifically, we examine two missing data scenarios for the Department of Energy Atmospheric Radiation Measurement
(DOE-ARM) program's Ka-band Zenith Radar (KaZR). This instrument collects cloud and precipitation information in a
vertical profile as weather passes over the radar and is used to generate time vs. height plots that are frequently used for
atmospheric research. The first scenario is a simulated instrument failure, where data is unavailable for up to several minutes.
The second scenario is a low-level blind zone. The low level blind zone is of particular relevance because the KaZR operates
with a burst and a pulse-compressed linear frequency modulated chirped pulse mode. When operating in chirped pulse mode,
data in the lower range gates is unavailable due to a receiver protection blanking region due to the longer pulse length (Widener
et al., 2012). Even the short burst pulse has a blind region near the surface based on the pulse width of the radar. Low-level
blind zones are also relevant to space-borne precipitation radars like the Tropical Rainfall Meteorology Mission (TRMM) and
the Global Precipitation Measurement mission (GPM) instruments, which can be blind at lower levels due to surface echos and
beam attenuation (Manabe and Ihara, 1988; Tagawa and Okamoto, 2003).

Robust methods for inpainting missing radar data have many possible uses. Accurately inpainting can provide more useful
operational meteorology products for dissemination to the public (Zhang et al., 2011) or for use in nowcasting (Prudden et al.,
2020; Agrawal et al., 2019) or aviation (Veillette et al., 2018). Furthermore, research applications often involve sophisticated,
high-level processing of radar data: for feature detection and tracking for instance (Feng et al., 2018). Producing radar products
for research purposes where missing and low quality data regions have been repaired could significantly accelerate research
projects by reducing or eliminating the need for researchers to develop their own code for error handling and data quality
issues. Ideally, an inpainting scheme for radar data should produce results that are accurate at the pixel level, but also visually
appealing, physically consistent, and plausible.

Image inpainting has long been an area of research in the fields of computer vision and image processing. The image inpaint-
ing problem involves restoring a missing or damaged region in an image by filling it with plausible data. Common applications
include digital photo editing, restoration of damaged photographs, restoring lost information during image compression and
transmission, etc. and many approaches with a range of application-specific advantages and varying levels of complexity exist
(Jam et al., 2021). Image inpainting schemes can be broken into several categories: texture synthesis based approaches assume
self-similarity in images and copy textures found in the un-damaged region of the image into the missing-data region (Efros and





Leung, 1999). Structure-based methods seek to extend large-scale structures into the missing data region from its boundaries and often focus on isophotes (lines of constant pixel intensity) that intersect the boundary (Criminisi et al., 2004). Diffusion based methods diffuse boundary or isophote information through the missing data region, typically by solving a partial differential equation within the region: the Laplace equation (Bertalmio et al., 2000) or the Navier-Stokes equations (Bertalmio et al., 2001) for instance. There are also sparse-representation and multi-resolutional methods that are typically geared towards

inpainting specific image classes, pictures of faces for example (Shih et al., 2003). Finally, there are many mixed approaches, like that used by Bugeau et al. (2010), that combine concepts of from two or more of these categories. Image inpainting is a large sub-field of image processing and for a more detailed overview see Jam et al. (2021) or Guillemot and Le Meur (2014).

In recent years, deep Convolutional Neural Networks (CNNs) have revolutionized the area of image inpainting research. The earliest applications of CNNs to the image inpainting problem involved using autoencoders that optimized pixel-level

loss (Jain and Seung, 2008). Research in this area began to quickly accelerate after the introduction of Generative Adversarial Networks (GANs) for image processing and synthesis by Goodfellow et al. (2014) however. GANs allow CNN-based inpainting schemes to hallucinate plausible small-scale variability, including textures and sharp edges, in the inpainted regions. GAN-based algorithms can produce extremely realistic results, to the point that it may not be obvious that inpainting has been performed. GAN-based inpainting involves training two CNNs side-by-side. One is the inpainting/generator network, which

takes a damaged image as input and attempts to fill in missing data regions. The second is a discriminator network that takes either undamaged images or outputs from the generator and attempts to classify them as real or fake, typically optimizing a binary cross-entropy (Goodfellow et al., 2014) or Wasserstien (Arjovsky et al., 2017) loss function. (Pathak et al., 2016) first performed inpainting with a GAN, using a combination of $\ell_2$-loss that optimizes pixel-level errors and ensures that the inpainting CNN can reproduce large-scale structures (though they may be blurry), and an adversarial loss, enforced by a

discriminator network, that ensures the inpainted regions look realistic and forces the generator to produce realistic sharp features and small scale variability. (Yang et al., 2017) expanded on this by using a combination of adversarial loss and feature loss based on the internal activations of image classification CNNs (Ledig et al., 2017). There has been a significant amount of subsequent work focusing on altered loss functions, incorporating updated CNN architectures like U-Net (Ronneberger et al., 2015), and other applications of these CNN training paradigms. An example is the image-to-image translation introduced by

(Isola et al., 2017) whose used a combined $\ell_1$ and adversarial loss function that is used here. The research in this area is far too broad to include a comprehensive overview here, so please refer to Elharrouss et al. (2020) for a more in-depth review of deep learning based inpainting.

In this study, we experiment with applying state of the art deep learning based image inpainting schemes to fill in several types of missing data regions simulated for two of the ARM program radars. The majority of past image inpainting research

is heavily focused on restoring missing regions in photographs. As a result, GAN-based methods are heavily optimized to produce visually appealing and plausible results, are not necessarily good at reproducing the ground-truth image in terms of pixel-level accuracy. Therefore two CNN-based inpainting paradigms are investigated, one that optimizes only the pixel level mean absolute error ($\ell_1$-loss), and one that optimizes combined $\ell_1$ and adversarial loss.



## 2 Data

The data used here are from two US DOE Atmospheric Radiation Measurement (ARM) program radars that were deployed
as part of the Cloud, Aerosol, and Complex Terrain Interactions (CACTI) field campaign from October 2018 to April 2019
(Varble et al., 2018). The field campaign deployed a large suite of instruments near the Sierras de Córdoba mountain range
in Argentina, including multiple radars, lidars, imagers, rain gauges, and many others, with the primary goals of investigating
the influence of orography, surface fluxes, aerosols, and thermodynamics on boundary layer clouds and on the initiation and
development of convection. The radars were deployed in a region just east of the mountain range and were able to observe
frequent warm boundary layer cloud and a range of convective systems at various points in their lifetime.

### 2.1 KaZR

The Ka-band Zenith Radar (KaZR) is a 35GHz vertically pointing cloud radar that has been deployed at many of the ARM
sites around the world (Widener et al., 2012). It is a doppler radar that produces time vs height observations of cloud and
precipitation and operates in both a burst and a pulse-compressed linear frequency-modulated chirped pulse mode. Here, we
have used the quality controlled burst pulse mode reflectivity, mean doppler velocity, and spectrum width fields from the CACTI
field campaign from October 15th, 2018 to April 30th, 2019 (Hardin et al., 2020a). A minimum reflectivity of -10dBZ was used
as a mask and any reflectivity, velocity, or spectrum width observations in pixels lower than this threshold were ignored. The
CNN takes samples that have 256 time-observations and 256 vertical range gates as inputs. The radar has a sampling frequency
of 2s and vertical resolution of 30m and this corresponds to about 8.5min by 7.5km. Because there are frequent time-periods
when there is no cloud over the radar and training/evaluating the CNN on blank samples is not useful, time periods longer than
256 samples when there was no weather observed were removed from the data-set. Because there is some noise in the data and
samples without weather still contain some pixels with reflectivity greater than -10dBZ, "no weather" periods were defined as
periods that do not contain any 10x10-pixel (300m x 20s) block with all reflectivity values >-10dBZ. After this filtering, the
data-set is split into a training set containing the first 80% of the data and a test set containing the last 20% (with respect to
time). The KaZR operated continuously throughout the field campaign (Hardin et al., 2020b).

### 2.2 C-SAPR2

The C-band Scanning ARM Precipitation Radar 2 (C-SAPR2) is a 5.7GHz scanning precipitation radar (Hardin et al., 2020b).
Here, we have used reflectivity, radial velocity, and spectrum width data from Plan Position Indicator (PPI) scans. The data
has a 1deg azimuth and 100m range resolution. The PPI scans used were pre-processed using the "Taranis" software package
(Hardin et al., 2021). Taranis provides quality control and produces a suite of useful geophysical parameters using the radar's
dual-polarization observations. This dataset has not yet been made publicly available but will be in the near future. The C-
SAPR2 suffered a hardware failure in February 2019 and was no longer able to rotate in the azimuth, so PPI scans are only
available from October 15th, 2018 to March 2nd, 2019 (Hardin et al., 2020b). For the majority of the field campaign, the
C-SAPR2 scan strategy involved performing a series of PPI scans at consecutively increasing elevation angles, followed by



a vertical scan, followed by a series of range height indicator scans. The whole process takes about 15 minutes. PPI scans at subsequent elevations are often similar because they are observed in quick succession, so the training set used here contains only one sweep from each volume scan. The sweeps used to construct the inpainting dataset were selected randomly from the 5-sweeps in each volume scan that contained the most weather (most observations with reflectivity >0dBZ). Many of these

still did not contain weather, and inpainting empty scans is not useful, so ultimately 1500 scans that contained weather were retained. As with the KaZR data, the first 80% were used for training and the last 20% were used for testing. The samples were randomly rotated and flipped in the azimuth during training to increase diversity in the training set. Pixels with reflectivity below a threshold of 0dBZ were blanked out.

## 3   Methods

Two deep-learning approaches are demonstrated here to inpaint missing data regions. The first involves using a single CNN. Radar data are intentionally degraded by removing randomly sized chunks of data. The exact manner in which the data is handled depends on the case and more information is provided in Section 4. The CNN is tasked with taking the degraded radar data along with a mask indicating the position of the missing data as an input and minimizing the mean absolute pixel-wise error between its output and the original (non-degraded data). This referred to as "$\ell_1$-CNN" throughout the manuscript because

it optimizes the $\ell_1$ norm of difference between its outputs and ground-truth. The second approach involves training two CNNs adversarially: one that performs inpainting and one that discriminates between infilled radar data and ground truth radar data. The inpainting network attempts to minimize both the mean absolute pixel error and the likelihood that the discriminator labels its output as inpainted data. The inpainting network is provided an additional random noise seed as input and is allowed to hallucinate plausible small scale variability in its outputs. This CNN is referred to as a Conditional Generative Adversarial

Network ("CGAN") later in the manuscript. The exact neural networks and training procedure are described in more detail below.

### 3.1   Convolutional Neural Network

Inpainting is done with a Unet++ style CNN (Zhou et al., 2018). This is based on a previous neural network architecture called a "Unet" (Ronneberger et al., 2015). These CNNs map a 2-D gridded input to an output with the same dimensions, and were

originally developed for image segmentation tasks. A U-net is composed of a spatial down-sampling and a corresponding up-sampling branch. The down-sampling branch is composed of a series of "blocks" consisting of multiple convolutional layers and the input data undergoes 2x2 down-sampling as it passes through each block while the number of feature channels is increased. This process trades spatial information for feature information. The down-sampling branch is followed by an up-sampling branch that increases the spatial dimensions and decreases the feature dimension. A key aspect of the Unet

is that it also includes skip connections between these two branches, where the output from each down-sampling block is provided as additional input to the upsampling block with the corresponding spatial resolution. This makes these networks particularly good at combining large-scale contextual information with pixel-level information. The Unet++ extends this idea





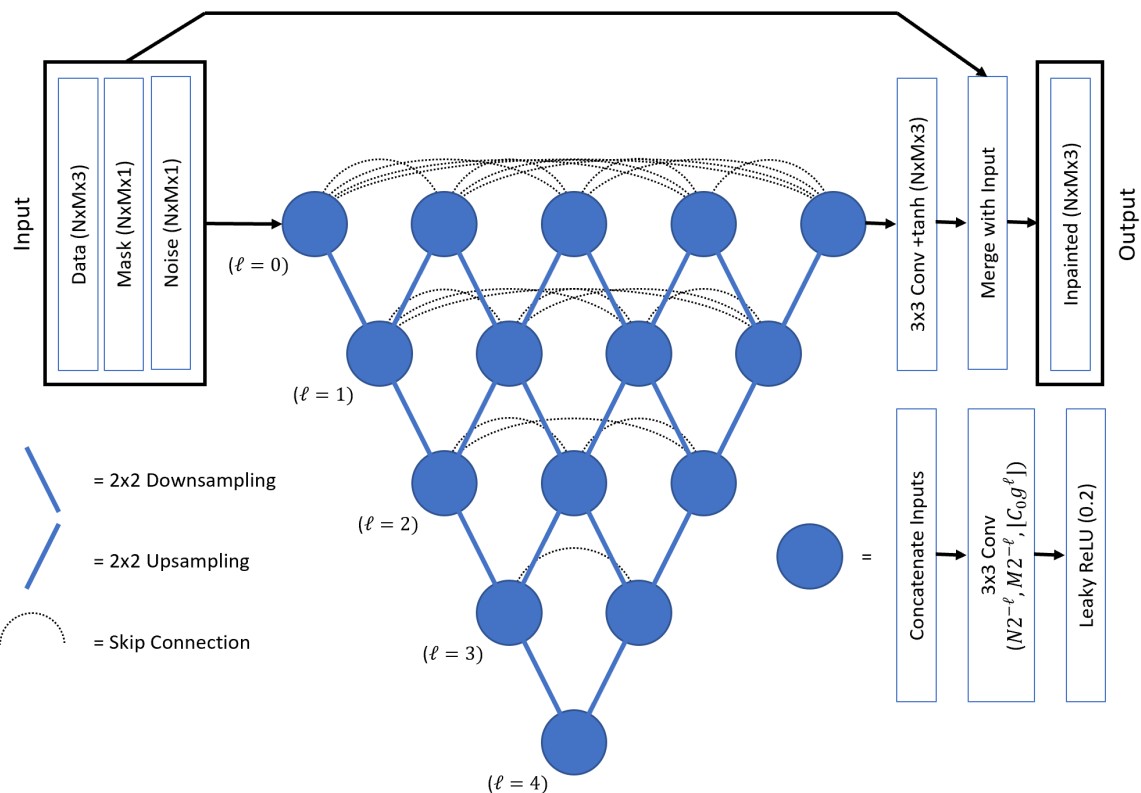

**Figure 1.** A diagram of the Unet++ (Zhou et al., 2018) style CNN used for inpainting. For the KaZR cases NxM = 256x256. For the C-SAPR2 cases NxM = 1024x128. $\ell$ represents the number of spatial down-sampling operations, $C_0$ is the number of output channels from the convolutional layers at the highest spatial resolution level, and $g$ determines the rate at which the number of channels increases for lower resolutions.

by constructing each skip-connection across the U-net from a set of densely connected (Huang et al., 2017) convolutional layers and including intermediate up- and down-sampling connections (the inclusion of intermediate down-sampling connections is a slight difference from the original Unet++ as described by Huang et al. (2017)). A diagram of the inpainting CNN is shown in Figure 1. The discriminator network consists of seven consecutive densely connected blocks (Huang et al., 2017) that consist of four convolutional layers with rectified linear unit transfer functions followed by 2x2 down-sampling and 0.1 dropout layers. The final output is a classification produced by a 1-neuron layer with a logistic (sigmoid) transfer function. A diagram of this CNN is included in the supplementary material.

## 3.2 Data Processing

Several pre- and post-processing operations need to be applied to the radar data so that it is suitable for use with the CNNs. Firstly, the data need to be standardized to a consistent input range of [-1,1]. The various fields used have different scales



and this ensures that they have similar weighting when computing loss. Furthermore, the CNN uses a $tanh$ transfer function after the last layer which ensures that the CNN outputs are limited to the same range as the inputs. Separate standardization

procedures were used for each of the fields. For reflectivity: data was clipped to a range of -10dBZ to 40 dBZ for KaZR data and a range of 0dBZ to 60 dBZ for C-SAPR2 data. In practice, we found that the $\ell_1$-CNN and many of the inpainting schemes that were used as benchmarks tended to smooth reflectivity values near cloud edges. Most cloud edges in the dataset involve a sharper gradient in reflectivity however, and this smoothing is a result of the inpainting/interpolation schemes struggling to correctly position the cloud edge. To mitigate this, the reflectivity data was linearly mapped to a range of [-0.5,1.0], and pixels

with reflectivity below the minimum threshold were assigned a value of -1.0, leaving a gap of 0.5 between clear and cloudy values. This gap helped the CNN and some of the diffusion- and interpolation-based inpainting schemes to produce sharp edges at the boundaries of precipitation and cloud regions. No such gap was used for the other two fields, but as a post-processing step, after inpainting, the other two fields were masked so that all locations with reflectivity outputs < -0.5 were considered clear pixels. For velocity, a de-aliasing scheme was first applied (see Section 3.3), then velocity values were linearly scaled

by a factor of $(8ms^{-1})^{-1}$ for the KaZR data and $(28ms^{-1})^{-1}$ for the C-SAPR2 data. Note that the instruments have Nyquist velocities of $8ms^{-1}$ and $16.5ms^{-1}$ respectively, and that the scanning radar is more likely to observe large velocities because the vertical component of velocity is typically smaller than the horizontal component for atmospheric motions. Finally, a $tanh$ function was applied to bound all the velocity inputs in a range of [-1,1]. Note that the unfolded velocities will often exceed the Nyquist velocity so the $tanh$ function was used to bound the transformed velocity data to [-1,1] instead of clipping. This non-

linearly compresses the high velocity values near $\pm 1$, but ensures that different large values remain unique and distinguishable from each other. The spectrum width data were clipped to a range of $[0, 2.5]ms^{-1}$ for KaZR and $[0, 5.5]ms^{-1}$ for C-SAPR2 and then linearly mapped to a range of [-1,1]. The inverse of each of these operations was performed to map the CNN outputs back to the range of the original dataset.

When degrading data for CNN training and testing, missing data regions for each field were created by setting all reflectivity

and spectrum width values in the region to -1 and all velocity values to 0. Large regions of clear pixels naturally exist in the training data of course, so an additional mask channel is provided as an input to the CNN to indicate the regions that need to be inpainted. The mask has values of 1 in missing data regions and 0 elsewhere. Because values outside the missing data region are known, there is no point in returning CNN outputs for these areas. The final operation performed by the CNN takes the output from the last convolutional layer (with a $tanh$ transfer function) and uses the mask to combine the known data from the

input with the CNN outputs in the missing data region. The exact operation is:

$$G(x) = mG'(x) + (1 - m)x \tag{1}$$

where $G(x)$ is the CNN output for input pixel $x$, $G'$ is the output from the last convolution + $tanh$ layer, and $m$ is the corresponding pixel in the mask. CNN outputs in initial experiments that used a mask with a sharp edge (transition from 1 to 0) around the inpainting region tended to contain noticeable artifacts at the edges. To help ensure that the features produced by

the CNN at the edges of the inpainting region matched with the features just outside of the region in the input data, a $n$-pixel buffer region was included where the values in the mask decrease linearly from 1 to 0. During training, $n$ is randomly selected



from a range of 1-17 to improve the robustness of the trained CNN. The result is that the final CNN outputs in this buffer region are a weighted average of the CNN output and the known input data. This significantly reduced artifacts near the edge of the inpainted region. Finally, in the CGAN case, an additional random seed was provided as a CNN input that allows the

CNN to hallucinate plausible small-scale variability. Here, this seed was included as an additional input channel containing random values sampled from a Gaussian distribution with a standard deviation of 0.5. We found that after training the CNNs generally did not did not rely on this random seed however, this is discussed in more detail in Section 5. In summary, the input channels to the inpainting CNN are: [-1,1]-standardized reflectivity, velocity, and spectrum width data, a [0,1] mask indicating the inpainting region with smoothed edges, and a channel of random seed data for the generator when training as a CGAN.

### 3.3 Doppler Velocity Folding

The doppler velocity data from both KaZR and C-SAPR2 contain velocity folding. Doppler radars can only unambiguously resolve radial velocities of plus or minus a maximum value known as the Nyquist velocity ($V_{max}$). $V_{max}$ is a function of the frequency and range resolution of the radar. Velocities that exceed the Nyquist velocity are mapped periodically back into this range, so that velocities slightly larger than $V_{max}$ are mapped to values slightly above (smaller magnitude) $-V_{max}$. The velocity

data used here has $V_{max} = 8ms^{-1}$ and $V_{max} = 16.5ms^{-1}$ for KaZR and C-SAPR2 respectively. Despite the smaller Nyquist velocity, the KaZR data are generally less susceptible to aliasing because vertical velocities in the atmosphere are typically smaller (in magnitude) than horizontal velocities. In practice, instances of velocity folding manifest as large jumps in velocity near the scale of $\pm 2V_{max}$, and because real-world meteorological conditions are extremely unlikely to cause jumps in velocity of this magnitude over such a small spatial scale, velocity folding is often easily detectable in contiguous cloud/precipitation

regions. Correcting folding is much more difficult than simply detecting it however. Many automated unfolding algorithms exist and this is still an active area of research.

We initially attempted to train the inpainting CNN on the velocity data without applying any unfolding scheme, but it struggled to adequately inpaint regions where folding had occurred. As noted above, the large jumps at the boundaries of aliased regions are extreme and non-physical, and while the CNN could reproduce large velocity-folded regions in its outputs,

it tended to smooth the change in velocity at the region boundaries over several pixels leading to a smoother transition and thus a result that most velocity un-folding algorithms that rely on detecting these jumps would fail on. We ultimately chose to implement a 2D flood-filling based de-aliasing algorithm that is usable for both the KaZR and C-SAPR2 data. The unfolding algorithm takes the velocity data, the Nyquist velocity, and a mask indicating clear pixels. For C-SAPR2 one sweep is processed at a time and for KaZR each net-CDF file retrieved from ARM is processed individually (typically about 20mins of data each,

though this is variable). The algorithm first breaks the velocity data into a set of contiguous regions that do not contain aliasing. This is done by first detecting the edges of regions with non-aliased velocity data by flagging all pixels that have velocity data where there is either a jump in velocity between that pixel and a neighboring pixel that exceeds $1.1V_{max}$ or there is a neighboring pixel that does not have velocity data. These region edge pixels are then used as seed points for a flood-fill algorithm which is applied iteratively until no seed points remain (every time a region is filled all seed points contained in that region get removed from the list). The regions are then processed from largest to smallest: if a region has no neighbors



its velocity remains unaltered and it is removed from the list of regions, if it does have neighbors, the largest neighboring region is identified and the mean change in velocity across the border between the two is used to correct the smaller region's velocity by adding or subtracting the appropriate multiple of $2V_{max}$. The smaller of the two regions is then integrated into the larger. This process continues until the list of contiguous velocity regions is emptied. This approach does have some failure

modes, typically associated with contiguous regions of aliased velocity that do not have any neighboring regions. We note that many other dealiasing schemes exist (Johnson et al. (2020) provide a KaZR specific scheme for instance) and may be worth investigating for future work, but this approach was sufficient for the CNNs trained here.

### 3.4 Training

The neural networks were trained using two different loss functions. In the first case, they were trained using a pixel-level

Mean Absolute Error (MAE) loss, also known as $\ell_1$ loss. This can be written as:

$$\mathcal{L} = \mathbb{E}_{x,y}[|y - G(x)|] \tag{2}$$

Here, $y$ is the true pixel value and $G(x)$ is the CNN output pixel value. We chose to limit the pixel-level loss so that it is only computed on pixels that are part of the infilled region (and the buffer pixels surrounding it, see Section 3.2) because the CNN is constructed to exactly reproduce the pixel values in the region with good data and because we used different sized missing

data regions during training and wanted them to have equal weighting when computing gradients and batch loss. The CNNs trained with MAE loss tend to produce more conservative results within the inpainted regions than CGANs (fewer details and extreme pixel values), but they are still particularly good and localizing and preserving sharp edges and larger structures and can outperform conventional inpainting and interpolation methods. In initial experiments mean squared error or $\ell_2$ loss was used but this led to extremely smoothed features in the inpainted region. We also trained CNNs as Conditional Generative

Adversarial Networks (CGANs), using a combination of $\ell_1$ and adversarial loss:

$$\mathcal{L}_G = \lambda \mathbb{E}_{x,y,z}[|y - G(x,z)|] - \mathbb{E}_{x,z}[\log(D(x, G(x,z)))] \tag{3}$$

$$\mathcal{L}_D = -\mathbb{E}_{x,y}[\log(D(x,y))] - \mathbb{E}_{x,z}[\log(1 - D(x, G(x,z)))] \tag{4}$$

Where $\mathcal{L}_G$ and $\mathcal{L}_D$ are the generator and discriminator losses respectively, $x$ is the input radar data and mask, $z$ is the random

seed input, $y$ is the ground truth data, $G(x,z)$ is the generator output, $D$ is the discriminator classification, and $\lambda$ is a constant used to weight the MAE and adversarial components of the generator loss. Refer to (Goodfellow et al., 2014) for a description of the adversarial loss and (Isola et al., 2017) for more discussion of the conditional adversarial loss function in Eqns. 3 and 4.

An Adam optimizer was used for training with: $\beta_1 = 0.9$, $\beta_2 = 0.999$, and $\epsilon = 10^{-7}$ for the $\ell_1$ case and $\beta_1 = 0.5$ for the

CGAN case. Other training details depend on the scenario and are summarized in Table 1. Table 1 shows specific information





| Case | Loss | $N \times M$ | $C_0$ | $g$ | Depth | Batch Size | Training Batches | LR-Reduction at Batches |
|------|------|-----------|-------|-----|-------|------------|------------------|-------------------------|
| KaZR Outage | $\ell_1$ | $256 \times 256$ | 8 | 2 | 5 | 8 | $5 \times 10^5$ | $4 \times 10^5$, $4.75 \times 10^5$ |
| KaZR Outage | Eq.3 | $256 \times 256$ | 14 | 1.75 | 5 | 16 | $1.6 \times 10^5$ | $0.3 \times 10^5$, $1.25 \times 10^5$ |
| KaZR Blind Zone | $\ell_1$ | $256 \times 256$ | 8 | 2 | 5 | 8 | $5 \times 10^5$ | $4 \times 10^5$, $4.75 \times 10^5$ |
| KaZR Blind Zone | Eq.3 | $256 \times 256$ | 14 | 1.75 | 5 | 16 | $1.3 \times 10^5$ | $0.7 \times 10^5$, $1.1 \times 10^5$ |
| C-SAPR2 Blockage | $\ell_1$ | $1024 \times 128$ | 10 | 2 | 6 | 8 | $5 \times 10^5$ | $4 \times 10^5$, $4.75 \times 10^5$ |
| C-SAPR2 Blockage | Eq. 3 | $1024 \times 128$ | 12 | 1.75 | 6 | 16 | $6.5 \times 10^4$ | $1.9 \times 10^4$ |

**Table 1.** Table of neural network hyper-parameters and training parameters. $N \times M$, $C_0$, $g$, and "Depth" are hyper-parameters that define the size and shape of the CNN and are shown in Figure 1 ("Depth" refers to the number of down-sampling operations or the maximum value of $\ell$ from Figure 1). "Batch Size," "Training Batches," and "LR-Reduction at Batches" refer to the number of samples per mini-batch, the total number of mini-batches/weight updates during training, and the batch number after which a learning rate reduction was performed (respectively).

about the CNN hyper-parameters, the batch size and number of batches used during training, and when the learning rate was decreased during training for each of the inpainting scenarios. For the $\ell_1$ cases, an initial learning rate of $5 \times 10^{-4}$ was used and was reduced by a factor of 10 twice during training after a set number of batches. For the CGAN cases, an initial learning rate of $1 \times 10^{-4}$ was used for the generator network and a learning rate of $1.5 \times 10^{-4}$ was used for the discriminator network. These were both manually reduced by a factor of 10 during training based on monitoring the adversarial loss and sample outputs for several randomly selected cases from the training set.

## 4 Results

In this section, the results of the CNN-based inpainting are compared to several common inpainting techniques. We first introduce these benchmark schemes and then discuss the error metrics used. Finally, results for each of the three inpainting scenarios are discussed separately. For space, only one sample case is shown for each of the inpainting scenarios, but many more have been made available online [1]. The examples shown in the manuscript were chosen blindly but not randomly, meaning we picked cases to include based on the ground truth but without consulting the CNN output. This was to ensure that the examples included a sufficient amount of cloud/precipitation to be of interest.

### 4.1 Benchmark Inpainting Schemes

CNN output was compared to several more conventional inpainting schemes of varying complexity. Examples of each of these schemes applied to the KaZR inpainting scenarios are shown in Figure 2. The same pre- and post-processing used for the inpainting CNNs is used for the inpainting schemes, as described in Section 3.2. We found that, in practice, this also helped

---

[1] https://drive.google.com/file/d/1UysTazBovGcxm1Sl_Q8xDU_BIrRhkZPX/view?usp=sharing





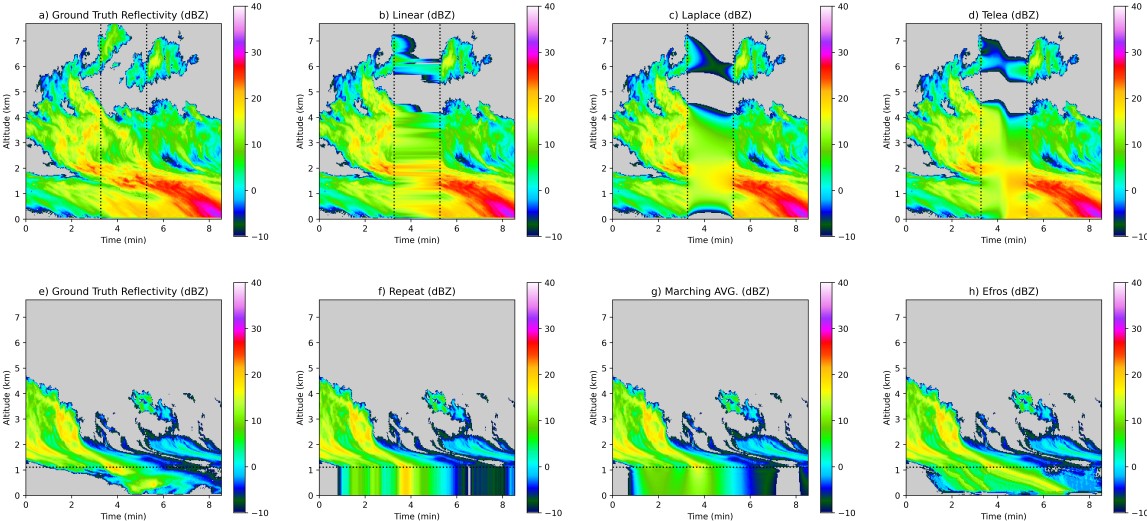

**Figure 2.** Example outputs from each of the benchmark inpainting schemes applied to KaZR reflectivity data. Panels b-d demonstrate the schemes used for the KaZR outage and C-SAPR2 beam blockage scenarios. Panels f-h show the schemes used for the low-level blind zone scenario. Ground truth is shown in panels a and e (left column).

the inpainting schemes generate sharper borders near cloud edges. Because the KaZR low-level blind zone scenario only has a single boundary with information that can be used for inpainting (the upper boundary), a different set of benchmark schemes were used for this case that are applicable to this type of scenario. The first three inpainting schemes below were used for both the KaZR outage and C-SAPR2 beam blockage scenarios which have 2-3 boundaries with information, while the last three were used for low-level inpainting.

**Linear Interpolation:** This is simple 1-dimensional linear interpolation between opposite boundaries of the missing data region. For the KaZR data, it is done with respect to time, and for the C-SAPR2 data it is done with respect to azimuth angle. Typically this approach performs well in terms of MAE but produces unrealistic results. It also does not take into account variability with respect to height (range) for KaZR (C-SAPR2).

**Laplace:** This scheme involves numerically solving the 2-dimensional Laplace equation on the interior of the missing data region. Firstly, we use linear interpolation to define the values on any missing boundaries (in the KaZR outage case, we interpolate the values for the bottom and top range gates for instance). Then:

$$0 = \alpha \nabla^2 z \qquad (5)$$

is solved numerically on the interior of the inpainting region. Here, $z$ is the field being inpainted, and $\nabla$ is the Laplacian with respect to time/height for KaZR or azimuth/range for C-SAPR2, and $\alpha = \frac{1}{50}$ and represents the diffusivity. The solution is found to a tolerance of 0.0001 using an explicit forward differencing scheme with a nine-point stencil, which was suffi-





cient for this application. This method produces particularly smoothed results in the missing data regions, though it is better than other methods at accurately producing large 2D structures (for instance linear interpolation only considers 1-D variability).

**Telea** The Telea inpainting scheme (Telea, 2004) is a fast inpainting algorithm that involves marching inward from the boundary of a missing data region and combines concepts of isophote-based inpainting and 2-D averaging-based diffusion. We used the implementation in the Python CV2 package with a radius of 16 pixels. This approach also produces fairly smoothed results and cannot produce small-scale variability like turbulence in the missing data regions, but does a better job of extending larger sharp features like cloud edges than the "Laplace" method.

**Repeat:** This is the simplest inpainting scheme used here and involves simply repeating the data at the upper boundary of the low-level blind zone down to the lowest range gate.

**Marching Average** Each horizontal line in the missing data region is inpainted from the uppermost level downwards. The assigned values are the average of the above data within 4 pixels. This method is conceptually similar to downwards repetition of the boundary data, but generates a smoothed result.

**Efros and Leung:** Efros and Leung (1999) present a texture synthesis model for image extrapolation that is well suited for inpainting missing data regions on the edges of images. Here, the process also involves marching from the highest missing data level downwards and filling in the missing data region one level at a time. First, a dictionary of exemplars is built by sampling 5x5-pixel patches in the area up to 64 pixels (192m) above the missing data region with a stride of 2 pixels. Each pixel is filled by computing the pixel-wise mean squared error between the surrounding 5x5 patch (not considering pixels yet to be inpainted) and the dictionary, and filling the missing pixel with the value of the center pixel from the closest matching patch. This scheme is designed to generate realistic textures in the inpainted region.

## 4.2 Error Metrics

Three error metrics are used to evaluate the inpainting results. They have been selected to provide a comprehensive evaluation of the quality of the outputs that considers the quality of the spatial variability and distribution of values generated by the inpainting schemes in addition to pixel-level error.

**Mean Absolute Error:** The Mean Absolute Error (MAE) is the primary error metric used. Panels a-c in Figures 4, 6, and 8, show MAE for the three fields analyzed. MAE was chosen because it is simple, the values are dimensional and easy to interpret, and it can be used directly in the loss function of the CNNs. Past CNN-based image inpainting work (Isola et al., 2017) has noted that MAE is preferable to other pixel-level losses because it is better for preserving sharp edges and small-scale variability. The MAE results for the test set are dimensionalized, meaning absolute error for velocities is shown in units of $ms^{-1}$ for instance, but it is important to note that because these errors are computed as an average over the missing data





region they are dependent on the amount of cloud present in each sample, and the inpainting scheme's ability to localize cloud edges (including many samples with little to no cloud in the test set would dramatically reduce MAE because inpainting no-data regions is trivial for all of the schemes). Because of this, the reported MAEs are mostly useful as a relative estimate of the skill of the different inpainting schemes as opposed to an absolute estimate of the difference between output reflectivity and ground truth for a particular sample for instance. In practice, MAE is particularly good for estimating the accuracy of an algorithm's ability to estimate the large-scale features in the missing data regions, but is not a good estimator of the plausibility of the output. Indeed, while the $\ell_1$-CNN is optimized on MAE and outperforms all other schemes under this metric, it produces smooth-looking outputs, and so two other error metrics are used to evaluate the accuracy of the small-scale variability and the distribution of the inpainted data.

**Earth Movers Distance:** The Earth Mover's Distance (EMD), also known as the Wasserstein metric, measures the similarity between probability density functions. Here, it is used to estimate the similarity of the distribution of values in the inpainting scheme outputs and the ground truth. The EMD imagines PDFs as piles of "dirt" and computes the "work" necessary to transform one distribution into another: the product of the amount moved and the distance. For 1-dimensional distributions this is simply the integral of the absolute difference between the PDFs. There are some important notes about our usage here: Firstly, no-data regions are included in the computation. For reflectivity and spectrum width they are assigned the minimum value for the field and for velocity they are assigned to $0 ms^{-1}$. This was necessary because the inpainting schemes do not necessarily produce cloud and precipitation features in the inpainted region of the exact same size (number of pixels) as the ground truth, but the area under the two PDFs that the EMD compares needs to be equal. Secondly, the EMD values are dimensional, but do not convey physical meaning unless compared to other EMD scores, so here we normalize the EMD and present it as the percentage of the worst possible EMD (lower values are better). Formally, the EMD is computed as:

$$EMD = \left( \frac{100}{z_{\max} - z_{\min}} \right) \int_{z_{\min}}^{z_{\max}} \left| \int_{z_{\min}}^{z} \left( Z_1(y) - Z_2(y) \right) dy \right| dz \tag{6}$$

where $z$ and $y$ integrate over the range of reflectivity values, $z_{\max}$ and $z_{\min}$ are the minimum and maximum reflectivity, and $Z$s represent PDFs of reflectivity data. EMD for the reflectivity field is shown panel d of Figures 4, 6, and 8. Plots of EMD for the other fields are included in the supplement.

**Power Spectral Density (PSD):** is used to estimate the ability of the inpainting schemes to produce plausible small-scale variability. This is an important error metric to consider because many inpainting approaches produce smooth results in the inpainted region that, while they may optimize pixel-level error reasonably well, are not representative of most atmospheric phenomena, which often contain small-scale turbulent features. Because some of the schemes behave differently along the different dimensions (time/height or range/azimuth), we separately compute PSD in each dimension. Here, the PSD is computed as:

$$PSD = 10 \log_{10} \left( \overline{\left( \mathcal{F}_x\{z\} \right)_y} \right) \tag{7}$$



where $\mathcal{F}_x$ is the Fast Fourier Transform with respect to dimension $x$, $\overline{(\cdot)}_y$ is the mean with respect to dimensions $y$, and $z$ is the reflectivity. For horizontal reflectivity PSD computed on the KaZR blind-zone case for instance, $z$ is reflectivity, $x$ is time, and

$y$ is height. In the following plots of PSD, the ground truth PSD is shown as a black line, and ideally the inpainting schemes yield PSD curves close to this black line. When viewing the results for this error metric it is important to note that some of the simplest techniques: linear interpolation and repeating boundary conditions, appear to produce realistic power spectra. They are only able to do so in one dimension however, because they are simply copying the realistic frequency information from the boundary into the missing data region. Along the other dimension, they do not produce any useful small-scale variability.

As with the EMD, the PSD metric cannot handle no-data regions well, so these regions are filled with the same default values described for EMD before computing the PSD. In Figures 4, 6, and 8 mean PSD for the reflectivity field is shown in panels e and f. Plots of the PSD for velocity and spectrum width are included in the supplementary material.

### 4.3   KaZR Outage Scenario

In this scenario, we simulate bad or missing data for a 16s to 168s period. While such an outage is less common than the

blind-zone or blockage scenario, it is possible; particularly when radars multi-task different modes and are not continuously in a zenith mode, but revisit it regularly. The outage case also provides an example of how the CNN-based inpainting schemes behave when there is information available on two boundaries. The CNN outputs for a sample case drawn from the test set are shown in Figure 3. This sample shows a 1-minute missing data period denoted by the vertical dashed lines and a 16-second buffer period is used before and after the missing data where the CNN outputs are blended with the ground truth data to

avoid edge artifacts. This scenario (Figure 3 a-c) was chosen because it contained complex cloud and precipitation features, including multi-layer cloud, areas of heavy precipitation, and many regions with turbulent motion. The outputs from the $\ell_1$-CNN are shown in Figure 3 panels d-f. They are clearly smoothed, and qualitatively speaking, if the markers to denote the outage period were not shown it would be obvious that inpainting had been done. This is particularly evident for the cloud near the top of the sample (around 6km). In the ground truth (panels a-c), this cloud does not extend through the missing data region,

but this is not made obvious by the information on the boundaries. The CNN has clearly struggled to localize the edges for this cloud. Nonetheless, the CNN outputs are considerably more appealing than the outputs from any of the baseline inpainting schemes. The $\ell_1$-CNN is particularly good at extending diagonal features like fall streaks across the missing data region and identifying an appropriate location for the upper boundary of the lower cloud. While some smoothing of the data has occurred, the results are much sharper than those achieved with the other schemes.

The improvement over conventional inpainting schemes is clear in Figure 4 panels a-c, where pixel-wise MAE is shown for each of the fields as a function of the duration of the missing data period. The $\ell_1$-optimizing CNN is the best performing scheme for each field. It also produces the most reasonable histogram of reflectivity outputs according to the EMD metric shown in Figure 4 panel d. The margin of improvement for these two metrics increases with the size of the missing data region. Interestingly, the linear interpolation approach, which is one of the least sophisticated inpainting schemes, performs the

second best in most cases. The vertical and horizontal PSD plots shown in Figure 4 panels e and f demonstrate the L1-CNN's (and other inpainting schemes') major limitation however: the tendency to over-smooth outputs. In panels e-f, the black curves







**Figure 3.** An example of inpainting a KaZR outage/missing data period. Panels a-c: ground truth, panels d-f: conventional CNN inpainting, panels g-i: CGAN inpainting.

represent the ground truth, and proximity to the black curve is better. The inpainting schemes typically have lower PSD than the ground truth and this is the most evident at high-frequencies, meaning the inpainting methods do not produce a realistic amount of small-scale variability. The linear interpolation approach has a reasonable PSD curve in the vertical, but this is because it simply copies vertical frequency information from the boundaries of the missing data region. It is the worst performer in the horizontal (time) dimension.

The only approach that performs well in both the horizontal and vertical component is of PSD is the CGAN. The output from the CGAN (Figure 3 g-i) is significantly more realistic than the $\ell_1$-CNN, to the degree that it may not be obvious that inpainting was performed without close examination. This is because it has generated plausible small scale variability in



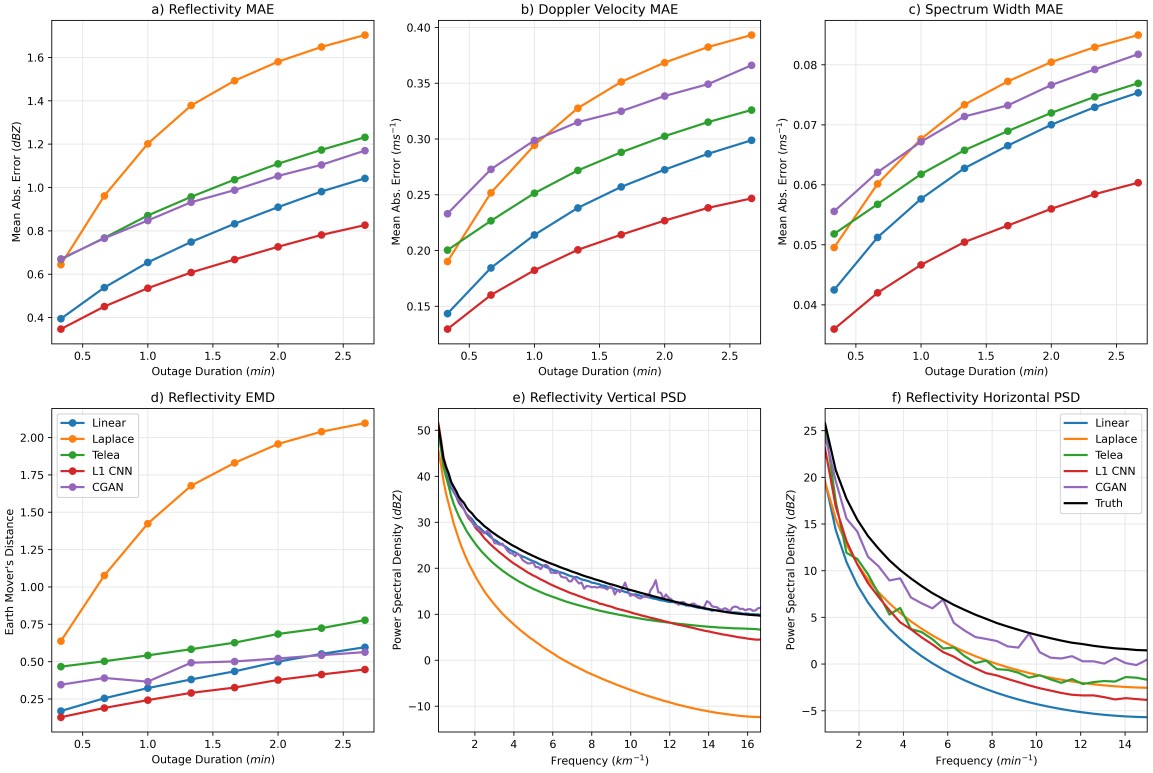

**Figure 4.** Error metrics computed on the KaZR outage inpainting scenario. Panels a-c show mean absolute pixel errors. Panel d shows the earth mover's distance. Panels e-f show power spectral density.

addition to inpainting the large scale features successfully. For instance, for the lower cloud/precipitation region in this sample, it has successfully generated a plausible and sharp upper boundary for the cloud and has extended large reflectivity features that span the missing data region across, much like the output from the $\ell_1$-CNN. In addition, it has added small-scale features, like jaggedness to the upper edge of the cloud, or turbulent motion visible in the vertical velocity field in the upper half of the cloud. The most noticeable unrealistic feature that is frequently produced by the CGAN is a tendency to introduce faint

linear structures and checkerboard patterns. This is a well documented problem for generative adversarial networks in general, and is a byproduct of the convolutions in the CNN (Odena et al., 2016). The results for the higher cloud are of particular interest. Here, the CGAN has opted to connect the cloud features on either side of the missing data region, and include positive velocities and high spectrum width indicating cloud-top turbulence. These features are not present in the ground truth data, but are realistic and consistent with cloud features commonly observed during the field campaign. Because the features are not an

exact match for the ground truth the scheme's MAE and EMD scores suffer, but they are still comparable to the scores of other inpainting schemes.



## 4.4 KaZR Blind Zone Scenario

**Figure 5.** An example of inpainting a KaZR low-level blind zone. Panels a-c: ground truth, panels d-f: conventional CNN inpainting, panels g-i: CGAN inpainting.

In this case, we simulate missing low-level KaZR data. This type of inpainting has multiple potential applications: Firstly, the KaZR operates in both a burst and chirped pulse mode. The chirped mode provides higher sensitivity, but does not retrieve usable near-surface data (Widener et al., 2012). Secondly, space-borne radars like GPM are unable to retrieve near-ground data due to interference from the surface (Tagawa and Okamoto, 2003) and the size of the blind zone depends on terrain and surface type. The CNN-based inpainting techniques presented here could easily be modified to work with such datasets. Here, KaZR burst pulse mode data is artificially degraded and all values below a randomly selected level from .21km - 2.01km are removed.



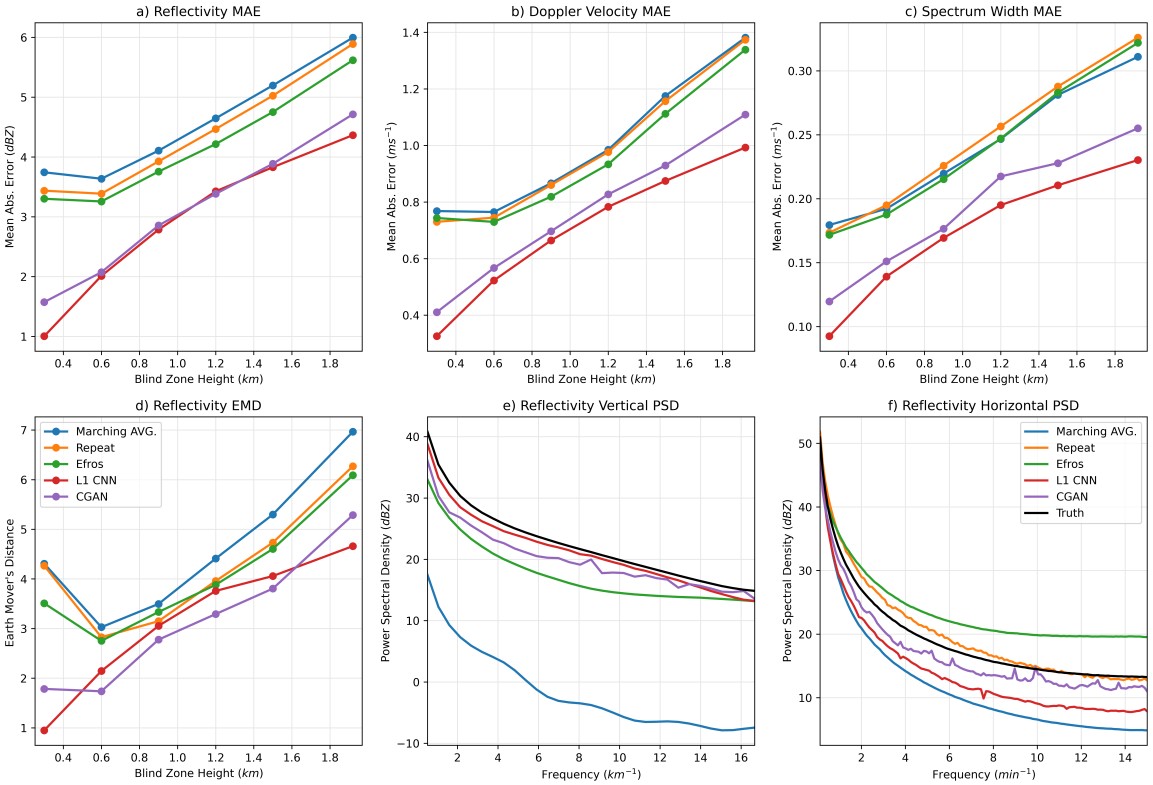

**Figure 6.** Error metrics computed on the KaZR low-level blind zone. Panels a-c show mean absolute pixel errors. Panel d shows the earth mover's distance. Panels e-f show power spectral density.

In this scenario, we also use a small buffer region (30-510m) near the top of the missing data region where observations are smoothly merged with the CNN output.

A sample from the test set is shown in Figure 5. The ground truth data (panels a-c) show a cloud (left side) with a fall streak extending diagonally downward towards the missing data region that weakens closer to the ground. Only about half of the missing data region contains cloud/precipitation, and so inpainting this particular case will involve accurately guessing the location of the cloud edge. The output from the $\ell_1$-CNN is shown in panels d-f. This scheme does a particularly good job of extending the large-scale features downwards diagonally into the missing data region, which is consistent with the ground truth. Note the leftmost cloud edge in the ground truth intersects the top of the missing data region at around 1min, but reaches the ground around 4.5min. The $\ell_1$-CNN does a particularly good job of capturing this. It does not however, introduce sharp features or small-scale variability. This is perhaps most noticeable in the spectrum width field, where values in the blind zone only range from about 0-0.5$ms^{-1}$. On the other hand, the CGAN does a much better job of including sharp and realistic features in the missing data region. Notably, in this case it chooses to increase the intensity of the main fall streak in the example all the way to





the surface, with correspond high values of reflectivity, negative velocity, and increased spectrum width. This is not consistent with the ground truth, but it is a plausible scenario.

The error metrics for the low-level inpainting case are shown in Figure 6. As in the simulated KaZR outage case discussed above, the classical CNN approach outperforms all of the baseline inpainting schemes in both MAE and EMD (panels a-d),

typically only having about 1/2-1/5 of the pixel-level error. This alone is a significant improvement and indicates that in the future, a deep-learning-based approach may be best for filling in missing low level radar data. Interestingly, in this case the CGAN performs about as well as the $\ell_1$-CNN in terms of the MAE and EMD. We hypothesize that this is due to the fact that there is only one boundary with information that can be used by the inpainting algorithms in this case. Compared to the KaZR outage case (Figure 4) the MAE for all of the algorithms has increased substantially in this case, though the MAE for the

CGAN less so. This may be because the other inpainting algorithms relied heavily on having 2-boundaries to accurately place the edges of inpainted clouds, and because the 2nd boundary is not available in this case, inability to correctly place cloud edges and large scale features becomes a large contributor to MAE. Meanwhile, a large contributor to the MAE of the CGAN is likely its tendency to generate small-scale pixel-level variability which would not necessarily be different between the two KaZR inpainting scenarios. At the same time, it performs comparably to the classical CNN when placing large-scale features.

Power spectral density is shown in panels e-f. Unsurprisingly, the CGAN (purple) is the scheme that consistently performs well for both horizontal and vertical components of PSD. Interestingly, the $\ell_1$-CNN performs well in the vertical component, but worse in the horizontal component. The "repeat" scheme does well in the horizontal component of PSD because it simply copies frequency information from the boundary of the missing data region, but cannot produce any variability in the vertical. Finally, the "Efros" scheme actually produces too much high-frequency variability in the horizontal component. This results

from the structure of the data-set. The scheme is simply copying textures from the area above the missing data region and the PSD reflects those textures. In conclusion, deep learning can considerably improve upon existing schemes for infilling missing low-level radar-data. Furthermore, for this specific application, there appears to be little reason not to use a CGAN-based approach that can generate plausible small-scale variability because it does not lead to significant increases in pixel-level error compared to the $\ell_1$-CNN.

## 4.5   Scanning Radar Beam Blockage Scenario

In this scenario, beam blockage is simulated using data from the C-SAPR2 scanning radar. Beam blockage due to nearby objects (like trees, buildings, or terrain) is a common problem for scanning radars, particularly at lower elevation angles. Of the three missing data scenarios used in this paper, this one is likely the most widely applicable due to the prevalance of scanning radars in operational systems. The ability to accurately fill in missing data due to beam blockage is useful for operational

weather radars, as it can provide more consistent inputs for weather models that ingest radar data, like nowcasting systems, and could be used to generate more appealing radar products for dissemination to the public. For the C-SAPR2 radar specifically, inpainting beam blockage areas would allow for easier application of high-level processing that might be used for research, such as feature detection and tracking. In particular, we would want an infilling system in these cases to accurately represent the distribution of the weather without straying too far from the ground truth. Furthermore, C-SAPR2 is deployed at ARM sites





**Figure 7.** An example of inpainting a C-SAPR2 beam blockage. Panels a-c: ground truth, panels d-f: conventional CNN inpainting, panels g-i: CGAN inpainting.

which may be more likely to suffer from beam blockage due to nearby objects because there is typically a large suite of other instruments deployed nearby and because clearing potential nearby blockages (tree removal or re-grading) or building a large structure to raise the antenna may not be an option. Missing data of this form can also be caused by attenuation of the radar wave due to strong precipitation, which is a common problem for C-band and higher frequency radars.

Here, we simulate beam blockages of 8-42 degrees starting anywhere from 1.6-25.6km from the radar. Again, a buffer of variable size from 1-17 pixels during training and 8-pixels during testing is used around the simulated blockage region to merge inpainted data and observations via Equation 1. Unlike the other two inpainting scenarios, this case has three boundaries with



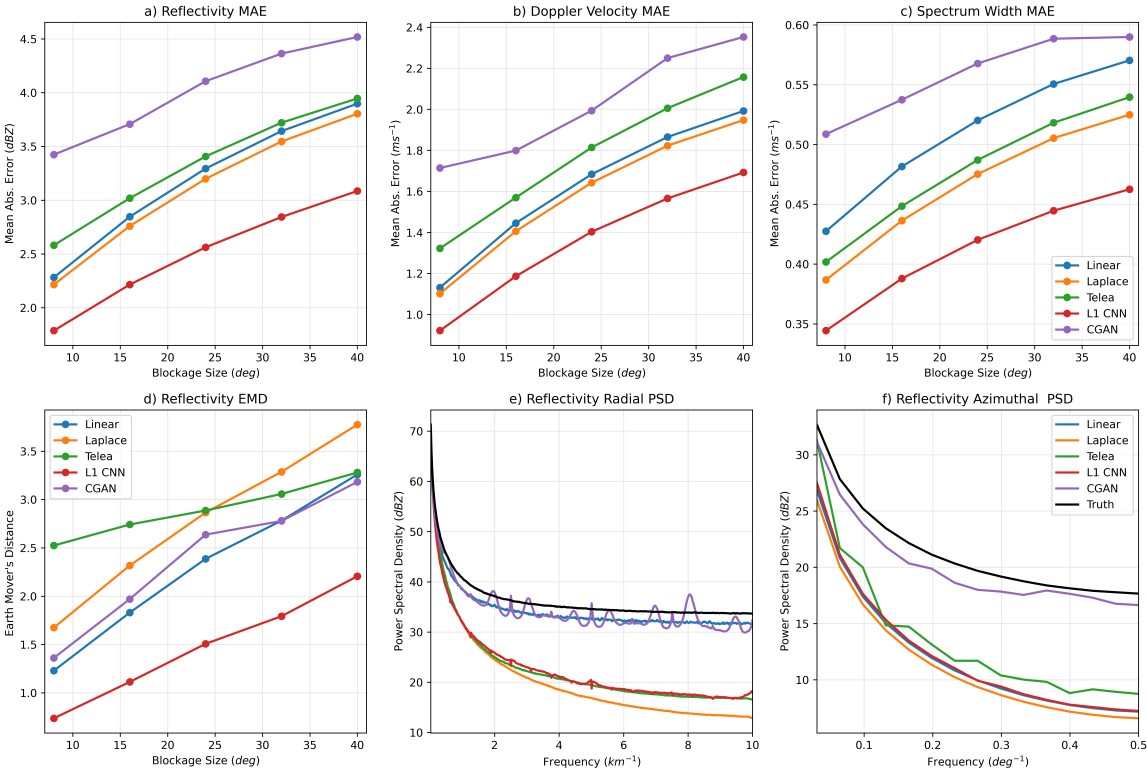

**Figure 8.** Error metrics computed on the C-SAPR2 beam blockage. Panels a-c show mean absolute pixel errors. Panel d shows the earth mover's distance. Panels e-f show power spectral density.

observations. The buffer at the corners where these boundaries meet is defined as the outer product of two vectors that linearly decrease from 1-0. The C-SAPR2 data used is defined in polar coordinates with 1-degree resolution in the azimuth and 100m range resolution. While it may be possible to inpaint the blockage region by applying a smaller CNN multiple times, it is

preferable to inpaint the entire blockage region at once to avoid introducing edge artifacts in the middle of the inpainted area. It is also desirable to perform inpainting at the native resolution of the radar data to avoid introducing artifacts or degrading the data quality when converting to a different coordinate system. Because of this, we chose to use a slightly different configuration of the CNN for the beam blockage case. The inputs are trimmed to 1024 (range gates) x 128 (degrees) so that the whole scan is not processed at once but the entire blockage region is. The location and the size of the blockages were chosen randomly

during training but the samples are all rotated in the azimuth so that the blockage appears in the center of the inputs to the CNNs.

This scenario differs from the KaZR inpainting scenarios because the CNN operates directly on the polar data. We made this choice so that the data did not have to be re-gridded, which would sacrifice resolution near the radar and could potentially introduce unwanted artifacts. Conventional CNNs may not be well-suited for operating on polar data however. This is because

the convolutional filters assume that the data is translationally invariant and that the features they learn are applicable every-



where in the image. The observed size (number of pixels occupied) of a weather feature of a fixed physical size can change drastically in polar coordinates depending on its distance from the antenna due to spread of the wave as it propagates from the radar however. This may mean that the CNNs have difficulty learning efficient physical representations of common weather features because physically similar features can appear quite different depending on their location. Put another way: CNNs

may require more filters to learn the same representations in polar coordinates than they would need to represent the same objects in Cartesian space. Nonetheless, training the CNN to process C-SAPR2 data went smoothly and the CNN learned to outperform conventional inpainting schemes by a comparable margin to the KaZR scenarios. Developing CNN architectures better suited to working in polar (or spherical) coordinate systems may be an important area of research in the future.

Example outputs of the beam blockage scenario are shown in Figure 7. The ground truth data are shown in panels a-c and

the dashed lines represent the region where data was removed to simulate beam blockage. This sample shows some diverse cloud and precipitation structure, with heavy precipitation occurring in the left portion of the missing data region and weaker precipitation on the right, in addition to a clear sky area and smaller precipitating feature closer to the radar. The $\ell_1$-optimizing CNN outputs are shown in panels d-f and the CGAN outputs are shown in panels g-i. Both neural networks do a good job of extending large scale features into the missing data region: the area of negative radial velocities near the left edge of the

blockage for instance. Again, the output from the $\ell_1$-CNN is too smooth, and it is obvious that inpainting has been performed. The CGAN introduces plausible small-scale variability in each of the fields however, and qualitatively speaking, the output looks realistic, to the point that it may be difficult to notice that a beam blockage has been filled in if it were not for the dashed lines in the figure. We have made other samples from the test set available for download. Compared to the KaZR scenarios, the CGAN has a much larger tendency to produce qualitatively low quality results when inpainting C-SAPR2 data. There are

many possible reasons for this, but the most likely seems to simply be the fact that the C-SAPR2 data is significantly different than that collected by KaZR and contains different types of structures that may be more difficult for the CNN to represent. Particularly notable are cases where the observed data are spatially smooth. In these cases, the CGAN has a tendency to introduce too much variability and sometimes edge artifacts in the inpainted region, and this makes it obvious that inpainting was performed. These cases are not common in the dataset however, and it appears that the CGAN did not learn to generalize

to them. Edge artifacts at the sides of the blockage region were more common in the C-SAPR2 case. Similar-looking radial lines are fairly common in the original C-SAPR2 data, and this may have made it difficult for the discriminator network to differentiate between naturally occurring radial features and edge artifacts.

As in the other two scenarios, the $\ell_1$-CNN significantly outperforms the baseline inpainting schemes in terms of both pixel-level MAE and EMD (Figure 8 panels a-d). The CGAN performs worse than the other schemes in terms of MAE, though this is

expected because it introduces small-scale variability that while plausible, is not necessarily in the correct location and inflates the MAE. On the other hand, the CGAN performs reasonably well in terms of EMD and dramatically outperforms all of the other schemes in reproducing the ground truth PSD (panels e-f). Again, note that the linear interpolation scheme only appears to perform well in terms of PSD when computed along the range dimension because it copies real variability from the edge of the blockage region. In panel e, there appear to be ringing artifacts in the CGAN PSD curve (purple). They are evenly spaced in

frequency and we hypothesize that they are related to weak checkerboard-like artifacts in the CNN output that result from the



convolutional filters. In summary, the CNNs are effective for inpainting beam blockage regions for C-SAPR2, and likely will be for other scanning radars. The $\ell_1$-optimizing CNN performs extremely well for pixel-level errors and the CGAN produces outputs with realistic power spectra, which may be preferable if a more visually appealing or physically plausible output is desired.

## 4.6 Importance of the Random Seed

After finishing training, we found that the outputs from the CGANs were not dependent on the random seed that was supplied with the inputs. Traditionally, generative adversarial networks are designed to take a random vector as an input that represents a latent space. The GANs use this random data to introduce variability in their outputs. All three of the CGANs trained here do not appear to use the random input however (changing the random data for a given inpainting case does not alter the output). Nonetheless, they are still able to produce a diverse set of outputs with plausible small-scale variability, which seems to imply that the CGANs leverage the natural variability present in the radar data in the unmasked part of the scan to generate diverse outputs. Furthermore, the small-scale content in the inpainted regions changes as the size of the blockage region (and hence what observations are available to the CNN) is changed, and is not a perfect match to ground truth, so the models have not over-fit. The authors of (Isola et al., 2017) make a similar observation. Rather than a random input they use dropout layers at both training and test time to introduce some variability in the generator network. They note that while using the dropout at test time does produce some additional variability in the outputs, it is less than expected. Developing inpainting schemes and CGANs that are appropriate for use with meteorological data that can adequately represent a range of variance in their outputs may be an important area of future research. When using CGANs for a task like now-casting for instance, the ability to quantify uncertainty in the CGAN outputs by generating multiple realizations would be valuable.

## 5 Discussion and Conclusions

In this work we demonstrated the capabilities of modern deep learning based inpainting schemes for filling in missing radar data regions. Two approaches were tested: a Convolutional Neural Network (CNN), that optimizes pixel-level mean absolute error, and a Conditional Generative Adversarial Network (CGAN) that is capable of generating realistic outputs. The CNNs were compared to conventional inpainting and interpolation schemes and, in general, the CNN-based inpainting can provide far superior results in terms of pixel-level error, the distribution of the output data, and their ability to generate realistic power spectra.

The inpainting results for the two types of CNN make clear that a trade-off exists between pixel level accuracy and physical realism. The $\ell_1$-CNN was able to outperform all other schemes in terms of pixel-level errors, but ultimately produced outputs that are smooth and are not representative of realistic atmospheric variability. On the other hand, realistic inpainting can be achieved using a CGAN approach, which can generate plausible cloud and precipitation structures to the degree that it may be difficult to notice that inpainting has been performed without a close inspection of the outputs. This is exemplified by the PSD curves computed on the CGAN output, which showed that CGANs can closely mimic the variability in the training data across





spatial scales. Ultimately, this trade-off between pixel-level accuracy and physical realism is a fundamental limitation of the inpainting problem: the true small-scale variability in a missing data region is not recoverable, and the problem of filling it in

is ill-posed because multiple physically plausible solutions exist. In other words, one must choose whether pixel-accuracy or realistic features are of greater importance given their task. An MAE of zero represents an exact match to the original data, and so some might argue that, for scientific datasets, pixel- or observation-level errors are a priority. An alternative view, however, is that in practice, the inpainted output from the $\ell_1$-CNN is always smoothed and unrealistic to the degree that we can say with near certainty that it is not representative of what actually occurred. The CGAN can at least provide a plausible result that is

unlikely to be true but cannot immediately be dismissed as incorrect. The choice to use a $\ell_1$- or CGAN style CNN for inpainting will ultimately be task dependent. For some applications, like extreme weather detection, a $\ell_1$-CNN is likely a better, more conservative, choice because it is unlikely to hallucinate an important feature. For generating visually appealing broadcast meteorology products or infilling blockages so that high-level processing can be applied in a research setting, a CGAN may be a better choice. In either case, our results demonstrate that CNN-based inpainting schemes can significantly outperform their

conventional counterparts for filling in missing or damaged radar data. Finally, while the capabilities of these schemes were demonstrated here on radar data, it should be noted that none of the CNN-based methods themselves utilize anything unique about radar and have significant potential for application to other instrument datastreams or even model data.

*Code and data availability.* The code used for this project is available from Github: https://github.com/avgeiss/radar_inpainting

Additional sample outputs for test-set cases have been made available via Google drive: https://drive.google.com/file/d/1UysTazBovGcxm1Sl_

Q8xDU_BIrRhkZPX/view?usp=sharing

The KaZR data is available from the Atmospheric Radiation Measurement program data discovery tool: https://adc.arm.gov/discovery/ and have filenames: "corkazrcfrgeqcM1.b1."

The CSAPR-2 data used in this study was processed using the Taranis software package which is currently in development and will be made publicly available in the near future (Hardin et al., 2021), some information can be found here: https://asr.science.energy.gov/meetings/stm/

presentations/2019/776.pdf, Accessed: 3-19-2021. The original CSAPR-2 data from the CACTI field campaign (without Taranis processing) can also be found using the ARM data discovery tool and have filenames: "corcsapr2cfrppiqcM1.b1."

*Author contributions.* Andrew Geiss performed experiments and wrote manuscript, Joseph C. Hardin conceived of idea and contributed to manuscript.

*Competing interests.* The authors declare no competing interests.



*Acknowledgements.* This work was funded by the Department of Energy's Atmospheric Radiation Measurement program. PNNL is operated for the Department of Energy by Battelle Memorial Institute under Contract DE-AC05-76RL01830.



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
