# Peer review of "Inpainting Radar Missing Data Regions with Deep Learning"

_Atmospheric Measurement Techniques, 2021_

## Author Response (AR2)

**Response to reviewers for: Inpainting Radar Missing Data Regions with Deep Learning**

**Andrew Geiss and Joseph C. Hardin**

Oct. 2021

Thank you to both reviewers and to the editor for your thoughtful comments. We sincerely appreciate you taking the time to review our work and provide constructive comments! We have used your feedback to improve the paper and have responded to each reviewer comment in order below:

**Reviewer 1**

130: Flips and rotations of images are used. Is there any way to quantify whether these add skill? Are there any downsides to flipping a meteorological image, thereby including physically impossible weather patterns in the training set?

This is a good point, and we think data augmentation deserves a more thorough treatment in the manuscript. We moved the discussion of data augmentation to the end of Section 3.4 ("Training" Lines 271-282) and expanded it. Many of the data augmentation schemes used in the deep learning literature were designed for use with images and, if applied to radar data, would result in weather features that are physically impossible, so we were careful here to only use augmentations that produce physically plausible samples. For KaZR data, flips were performed with respect to time (not in the vertical). For most cloud features that are embedded in the large-scale flow this results in a physically plausible sample and approximates large scale flow in the opposite direction. For weather features whose shape is heavily determined by the large-scale flow (e.g. fall streaks) this results in an unlikely but still very realistic looking sample. For C-SAPR2, we flipped only with respect to azimuth (not range) and performed rotations with respect to azimuth. Likewise, these augmentations still result in physically plausible radar data because they simulate an altered scan strategy or coordinate convention without changing the physical structure of the weather in the scan.

335: The performance on blank space is dismissed as trivial. What about the performance on a sparsely speckled background? In real time radar applications, masks can be leaky due to atmospheric and instrumental variations. The CGAN's creativity in hallucinating weather patterns could pose risks if the presence of speckle or other faint, spurious data could trigger inpainting of weather patterns that do not exist.

The expected use case for this involves scenarios where a blockage/outage/low-quality data is known to exist, and the inputs to the CNN require a pre-defined mask and setting the input fields to constant values in the region where inpainting is required, so we do not expect the algorithm to be used to inpaint speckle alone. This would be similar to fitting noise, so the pixel-level skill for any scheme would likely be low. There are cases in our test set where a missing data region contained a weather feature and speckle in different locations. In these cases, the conventional CNN is typically very conservative and does not interpolate speckle from the boundaries into the inpainting region. The CGAN will hallucinate similar speckle in the inpainting region to that near the boundary. We have attached a sample case from the KaZR dataset below, where there is some speckle that was not removed by the reflectivity threshold near the ground with a weather feature above.

410, 420, 455: The CGAN success should be discussed a little more. It has the poorest performance, underperforming all the non-NN techniques on certain metrics. Also, it's skill has risks as mentioned above. I know this is addressed a bit already, but I think the outstanding qualitative skill of the CGAN (while poor quantitative skill) requires a bit more discussion of the risks, how they might be addressed for applications, and some speculation on how to leverage the relative skills of the two novel techniques.

410, 420, 455: Additional discussion was added to section 5 (Lines 583-59).

**Editor**

**It's not clear to me how the potential overfitting issue was faced during training.**

The examples shown in Figures 3, 5, and 7 and the error metrics in Figures 4, 6, and 8 were computed on test sets that were held out during training, which indicates the models did not overfit. Because the radar data are temporally correlated, we selected the test sets to be contiguous chunks of observations from the end of the field campaign so information leakage between the training and test sets is unlikely. We noticed early on during this study that overfitting did not occur in general for the inpainting CNNs. This is likely due to a combination of the nature of the inpainting task, the size of the radar datasets used, and the size of the CNN. To overfit, the CNN would have to memorize the radar data in 2D. This corresponds to about  $7x10^8$  pixels for KaZR or  $6x10^8$  for C-SAPR2, but the CNN has only around  $5x10^6$  trainable parameters (depending on which CNN).

**Are the different inpainting techniques fully comparable? E.g., why the interpolation method is not applied in two dimensions? The precipitation field has intrinsic 2-d spatial correlation that cannot be easily reproduced by 1-d linear interpolation.**

They are all comparable in the sense that they all fill-in the missing data region and the same error metrics can be applied. We intentionally selected a diverse set of inpainting methods to compare the CNN to. You are correct that the 1-D scheme completely ignores the 2-D structure in the radar data though the Laplace, Telea, Marching Average, and Efros schemes all incorporate 2-D information. An interesting result is that the very simple 1-D interpolation and 1-D 'repeat' schemes performed comparably to the much more complex schemes in terms of pixel-level error. They are much worse qualitatively however. Finally, 2-D interpolation was not used because none of the inpainting scenarios had data on all four boundaries of the missing data region. In the case of beam blockage, the farthest ranges are always missing and so cannot constrain the 2D interpolation properly.

**Relatively simple correction techniques for beam-blockage due to orographic obstacles are available (e.g., Bech et al., 2003). It would be interesting to make a comparison with the proposed method.**

We agree. The existing techniques for correcting partial beam blockage typically rely on multiple sweeps or volume coverage patterns however, and here we focused on single sweeps (for C-SAPR) with complete blockage. The success on single sweeps in this study certainly suggests there is potential for handling partial beam blockage with a 3-D CNN using sweeps at multiple levels or a full volume scan which would be more comparable to the partial beam blockage correction methods in the literature. Additionally, this technique tends to be more general as it works on multiple modalities of data and does not necessarily rely on any expert derived techniques. As such, we expect it would likely work for things like lidars and profilers. We leave these as potential areas of future research however.

**Reviewer 2**

**It looks like all the test cases shown in the manuscript are masked manually. I wonder if the authors could include some real cases (with missing/damaged data) to illustrate the practical application performance.**

The scheme does not detect or mask low-quality data regions and the CNN expects that the data in the inpainting region has already been masked, so even in practical application an area that needs to be inpainted will need to be masked manually in a similar manner to the examples in the paper. There is no difference between manually masked good data used in the examples or manually masked low quality data because in either case all of the data in the inpainting region is completely removed from the inputs to the CNN before it is inpainted. To automate applying this, one could apply off the shelf blockage masks, but as we wanted to compare the quality of the infilling, and not the blockage masking, we went with manually blanked areas for now.

**Page 7, Line 175: The authors left a gap of 0.5 between clear and cloudy (reflectivity) values, and then use a reflectivity threshold to mask other two fields. I wonder if it is worth adopting a similar strategy for other two fields, instead of masking the output using reflectivity threshold. Please comment!**

P7 L175: We tried this exact approach early on during this study, adding a gap in the normalized velocity data between -0.25, 0, and 0.25 for instance, but the CNN performed worse. We opted not to include this experiment in the paper. It also makes sense to apply the same masks across all three fields to avoid showing spectrum width data where the reflectivity was deemed not to be significant for instance. For dual-polarimetric fields this becomes a more interesting problem as one could choose different masks, but that is beyond the current scope of research. An alternative approach that we have not tried might be to include a separate channel as output from the CNN that will act as a mask. Because masking based on the output reflectivity gave good results we did not explore this further.

**In Fig. 2, I do not really see any beam blockages. Am I missing anything?**

This figure is solely meant to provide a visual example of each of the inpainting schemes. Some of the inpainting schemes are appropriate for the low level blind zone scenario only (one boundary with data, bottom row of Figure 2), while the others are applicable to either the KaZR outage scenario or the C-SAPR2 beam blockage scenario (two-three boundaries with data). We opted to use a sample from KaZR to demonstrate what the benchmark inpainting schemes look like for the 2-boundary case so beam blockages are not shown in this figure but the same inpainting schemes from panels b-d are also used for the beam blockage cases. We have changed the caption to clarify.

**Page 21, Line 485-490: I understand that the data are trimmed to range gates by azimuth angles so they can be processed at the native resolution. But I do not quite understand why the data were trimmed to 128 degrees and then rotated (why not use the whole scan?). Please clarify!**

L485: This is due to computational limitations while training. The CNN has  $-5x10^6$  parameters and when training we use a minibatch of 8-16 samples, processing the entire scan will exhaust the VRAM on the GPU that was used for training. That said, even with more memory, it may be useful to only process fractions of a PPI scan because the trained CNN can then be applied to data taken during a partial sweep or during a sweep that does not have exactly 1-degree resolution in the azimuth. We have added a note about this on line 500.

Page 19, Paragraph 470: "prevalance" should be "prevalence"

Corrected